# LEARNING FAST SAMPLERS FOR DIFFUSION MODELS BY DIFFERENTIATING THROUGH SAMPLE QUALITY

**Daniel Watson**[*]**, William Chan, Jonathan Ho & Mohammad Norouzi**
Google Research, Brain Team
{watsondaniel,williamchan,jonathanho,mnorouzi}@google.com

## ABSTRACT

Diffusion models have emerged as an expressive family of generative models rivaling GANs in sample quality and autoregressive models in likelihood scores. Standard diffusion models typically require hundreds of forward passes through the model to generate a single high-fidelity sample. We introduce Differentiable Diffusion Sampler Search (DDSS): a method that optimizes fast samplers for any pre-trained diffusion model by differentiating through sample quality scores. We present Generalized Gaussian Diffusion Models (GGDM), a family of flexible non-Markovian samplers for diffusion models. We show that optimizing the degrees of freedom of GGDM samplers by maximizing sample quality scores via gradient descent leads to improved sample quality. Our optimization procedure backpropagates through the sampling process using the reparametrization trick and gradient rematerialization. DDSS achieves strong results on unconditional image generation across various datasets (*e.g.,* FID scores on LSUN church 128x128 of 11.6 with only 10 inference steps, and 4.82 with 20 steps, compared to 51.1 and 14.9 with strongest DDPM/DDIM baselines). Our method is compatible with any pre-trained diffusion model without fine-tuning or re-training required.

## 1 INTRODUCTION

Denoising Diffusion Probabilistic Models (DDPM) (Sohl-Dickstein et al., 2015; Song & Ermon, 2019; Ho et al., 2020) have emerged as a powerful family of generative models, capable of synthesizing high-quality images, audio, and 3D shapes (Ho et al., 2020; 2021; Chen et al., 2021a;b; Cai et al., 2020; Luo & Hu, 2021). Recent work (Dhariwal & Nichol, 2021; Ho et al., 2021) shows that DDPMs can outperform Generative Adversarial Networks (GAN) (Goodfellow et al., 2014; Brock et al., 2018) in generation quality, but unlike GANs, DDPMs admit likelihood computation and much more stable training dynamics (Arjovsky et al., 2017; Gulrajani et al., 2017).

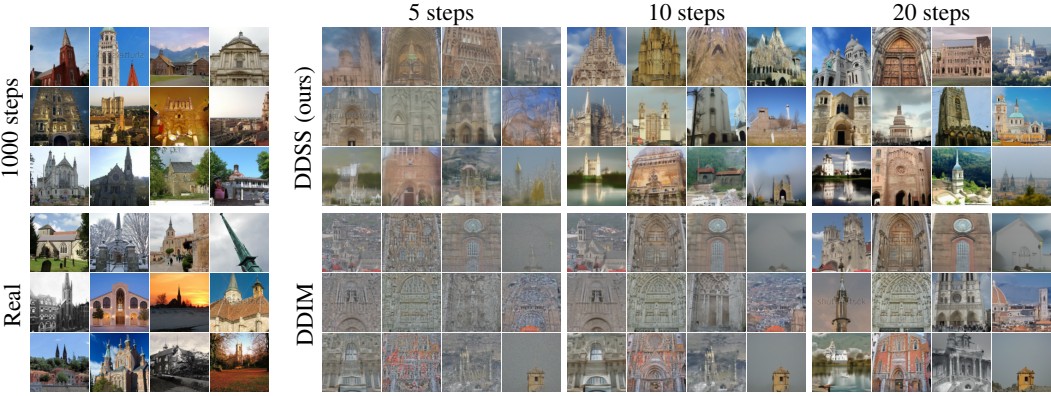

Figure 1: Non-cherrypicked samples from DDSS (ours) and strongest DDIM($\eta = 0$) baseline for unconditional DDPMs trained on LSUN churches 128×128. All samples are generated with the same random seed. Original DDPM samples (1000 steps) and training images are shown on the left.

---

[*]Work done as part of the Google AI Residency.

However, GANs are typically much more efficient than DDPMs at generation time, often requiring a single forward pass through the generator network, whereas DDPMs require hundreds of forward passes through a U-Net model. Instead of learning a generator directly, DDPMs learn to convert noisy data to less noisy data starting from pure noise, which leads to a wide variety of feasible strategies for sampling (Song et al., 2021b). In particular, at inference time, DDPMs allow controlling the number of forward passes (a.k.a. *inference steps*) through the denoising network (Song et al., 2020; Nichol & Dhariwal, 2021).

It has been shown both empirically and mathematically that, for any sufficiently good DDPM, more inference steps leads to better log-likelihood and sample quality (Nichol & Dhariwal, 2021; Kingma et al., 2021). In practice, the minimum number of inference steps to achieve competitive sample quality is highly problem-dependent, *e.g.,* depends on the complexity of the dataset, and the strength of the conditioning signal if the task is conditional. Given the importance of generation speed, recent work (Song et al., 2020; Chen et al., 2021a; Watson et al., 2021) has explored reducing the number of steps required for high quality sampling with pretrained diffusion models. See Section 7 for a more complete review of prior work on few-step sampling.

This paper treats the design of fast samplers for diffusion models as a differentiable optimization problem, and proposes *Differentiable Diffusion Sampler Search* (DDSS). Our key observation is that one can unroll the sampling chain of a diffusion model and use reparametrization trick (Kingma & Welling, 2013) and gradient rematerialization (Kumar et al., 2019a) to optimize over a class of parametric few-step samplers with respect to a global objective function. Our class of parameteric samplers, which we call Generalized Gaussian Diffusion Model (GGDM), includes Denoising Diffusion Implicit Models (DDIM) (Song et al., 2020) as a special case and is motivated by the success of DDIM on fast sampling of diffusion models.

An important challenge for fast DDPM sampling is the mismatch between the training objective (*e.g.,* ELBO or weighted ELBO) and sample quality. Prior work (Watson et al., 2021; Song et al., 2021a) finds that samplers that are optimal with respect to ELBO often lead to worse sample quality and Fréchet Inception Distance (FID) scores (Heusel et al., 2017), especially with few inference steps. We propose the use of a *perceptual* loss within the DDSS framework to find high-fidelity diffusion samplers, motivated by prior work showing that their optimization leads to solutions that correlate better with human perception of quality. We empirically find that using DDSS with the Kernel Inception Distance (KID) (Bińkowski et al., 2018) as the perceptual loss indeed leads to fast samplers with significantly better image quality than prior work (see Figure 1). Moreover, our method is robust to different choices of kernels for KID.

Our main contributions are as follows:

1. We propose Differentiable Diffusion Sampler Search (DDSS), which uses the reparametrization trick and gradient rematerialization to optimize over a parametric family of fast samplers for diffusion models.
2. We identify a parametric family of Generalized Gaussian Diffusion Model (GGDM) that admits high-fidelity fast samplers for diffusion models.
3. We show that using DDSS to optimize samplers by minimizing the Kernel Inception Distance leads to fast diffusion model samplers with state-of-the-art sample quality scores.

## 2 BACKGROUND ON DENOISING DIFFUSION IMPLICIT MODELS

We start with a brief review on DDPM (Ho et al., 2020) and DDIM (Song et al., 2020). DDPMs pre-specify a Markovian forward diffusion process, which gradually adds noise to data in $T$ steps. Following the notation of Ho et al. (2020),

$$q(\boldsymbol{x}_0, ..., \boldsymbol{x}_T) \quad = \quad q(\boldsymbol{x}_0) \prod_{t=1}^{T} q(\boldsymbol{x}_t|\boldsymbol{x}_{t-1}) \tag{1}$$

$$q(\boldsymbol{x}_t|\boldsymbol{x}_{t-1}) \quad = \quad \mathcal{N}(\boldsymbol{x}_t|\sqrt{1-\beta_t}\boldsymbol{x}_s, \beta_t\boldsymbol{I}), \tag{2}$$

where $q(\boldsymbol{x}_0)$ is the data distribution, and $\beta_t$ is the variance of Gaussian noise added at step $t$. To be able to gradually convert noise to data, DDPMs learn to invert (1) with a model $p_\theta(\boldsymbol{x}_{t-1}|\boldsymbol{x}_t)$, which

is trained by maximizing a (possibly reweighted) evidence lower bound (ELBO):

$$\mathbb{E}_q \left[ D_{\mathrm{KL}}[q(\boldsymbol{x}_T|\boldsymbol{x}_0)\|p(\boldsymbol{x}_T)] + \sum_{t=2}^{T} D_{\mathrm{KL}}[q(\boldsymbol{x}_{t-1}|\boldsymbol{x}_t,\boldsymbol{x}_0)\|p_\theta(\boldsymbol{x}_{t-1}|\boldsymbol{x}_t)] - \log p_\theta(\boldsymbol{x}_0|\boldsymbol{x}_1) \right] \quad (3)$$

DDPMs specifically choose the model to be parametrized as

$$
\begin{aligned}
p_\theta(\boldsymbol{x}_{t-1}|\boldsymbol{x}_t) &= q\left(\boldsymbol{x}_{t-1}\bigg|\boldsymbol{x}_t, \hat{\boldsymbol{x}}_0 = \frac{1}{\sqrt{\bar{\alpha}_t}}(\boldsymbol{x}_t - \sqrt{1-\bar{\alpha}_t}\boldsymbol{\epsilon}_\theta(\boldsymbol{x}_t,t))\right) \\
&= \mathcal{N}\left(\boldsymbol{x}_{t-1}\bigg|\frac{1}{\sqrt{1-\beta_t}}\left(\boldsymbol{x}_t - \frac{\beta_t}{\sqrt{1-\bar{\alpha}_t}}\boldsymbol{\epsilon}_\theta(\boldsymbol{x}_t,t)\right), \frac{1-\bar{\alpha}_t}{1-\bar{\alpha}_{t-1}}\beta_t\boldsymbol{I}_d\right)
\end{aligned}
\quad (4)
$$

where $\bar{\alpha}_t = \prod_{s=1}^{t}(1-\beta_t)$ for each $t$. With this parametrization, maximizing the ELBO is equivalent to minimizing a weighted sum of denoising score matching objectives (Vincent, 2011).

The seminal work of Song et al. (2020) presents Denoising Diffusion Implicit Models (DDIM): a family of evidence lower bounds (ELBOs) with corresponding forward diffusion processes and samplers. All of these ELBOs share the same marginals as DDPM, but allow arbitrary choices of posterior variances. Specifically, Song et al. (2020) note that is it possible to construct alternative ELBOs with only a subsequence of the original timesteps $S \subset \{1, ..., T\}$ that shares the same marginals as the construction above (i.e., $q_S(\boldsymbol{x}_t|\boldsymbol{x}_0) = q(\boldsymbol{x}_t|\boldsymbol{x}_0)$ for every $t \in S$, so $q_S$ defines a faster sampler compatible with the pre-trained model) by simply using the new contiguous timesteps in the equations above. They also show it is also possible to construct an *infinite* family of non-Markovian processes $\{q_\sigma : \sigma \in [0,1]^{T-1}\}$ where each $q_\sigma$ also shares the same marginals as the original forward process with:

$$q_\sigma(\boldsymbol{x}_0, ..., \boldsymbol{x}_t) = q(\boldsymbol{x}_0)q(\boldsymbol{x}_T|\boldsymbol{x}_0)\prod_{t=1}^{T-1} q_\sigma(\boldsymbol{x}_t|\boldsymbol{x}_{t+1}, \boldsymbol{x}_0) \quad (5)$$

and where the posteriors are defined as

$$q_\sigma(\boldsymbol{x}_{t-1}|\boldsymbol{x}_t, \boldsymbol{x}_0) = \mathcal{N}\left(\boldsymbol{x}_{t-1}\bigg|\sqrt{\bar{\alpha}_{t-1}}\boldsymbol{x}_0 + \sqrt{1-\bar{\alpha}_{t-1}-\sigma_t^2}\cdot\frac{\boldsymbol{x}_t - \sqrt{\bar{\alpha}_t}\boldsymbol{x}_0}{\sqrt{1-\bar{\alpha}_t}}, \sigma_t^2\boldsymbol{I}_d\right) \quad (6)$$

Song et al. (2020) empirically find that the extreme case of using all-zero variances (*a.k.a.* DDIM($\eta = 0$)) consistently helps with sample quality in the few-step regime. Combined with a good selection of timesteps to evaluate the modeled score function (*a.k.a. strides*), DDIM($\eta = 0$) establishes the current state-of-the-art for few-step diffusion model sampling with the smallest inference step budgets. Our key contribution that allows improving sample quality significantly by optimizing sampler families is constructing a family that generalizes DDIM. See Section 4 for a more complete treatment of our novel GGDM family.

## 3 DIFFERENTIABLE DIFFUSION SAMPLER SEARCH (DDSS)

We now describe DDSS, our approach to search for fast high-fidelity samplers with a limited budget of $K < T$ steps. Our key observation is that one can backpropagate through the sampling process of a diffusion model via the reparamterization trick (Kingma & Welling, 2013). Equipped with this, we can now use stochastic gradient descent to learn fast samplers by optimizing any given differentiable loss function over a minibatch of model samples.

We begin with a pre-trained DDPM and a family of $K$-step samplers that we wish to optimize for the given DDPM. We parametrize this family's degrees of freedom as simple transformations of trainable variables. We experiment with the following families in this paper, but emphasize that DDSS is applicable to any other family where model samples are differentiable with respect to the trainable variables:

- **DDIM**: we parametrize the posterior variances with $\sigma_t = \sqrt{1-\bar{\alpha}_{t-1}}\,\mathrm{sigmoid}(v_t)$, where $v_1, ..., v_K$ are trainable variables (the $\sqrt{1-\bar{\alpha}_{t-1}}$ constant ensures real-valued mean coefficients; see the square root in Equation 6).

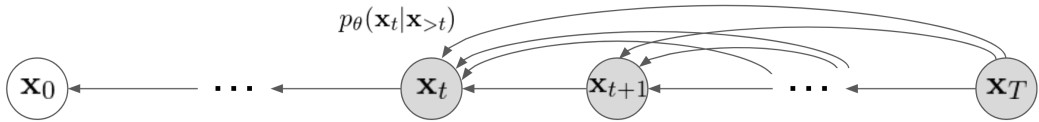

Figure 2: Illustration of GGDM. To improve sample quality, our novel family of samplers combines information from all previous (noisier) images at every denoising step.

- **VARS**: we parametrize the marginal variances of a DDPM as $\mathrm{cumsum}(\mathrm{softmax}([\boldsymbol{v}; 1]))_t$ instead of fixing them to $1 - \bar{\alpha}_t$. This ensures they are monotonically increasing with respect to $t$ (appending a one to ensure $K$ degrees of freedom).
- **GGDM**: a new family of non-Markovian samplers for diffusion models with more degrees of freedom illustrated in Figure 2 and defined in Section 4. We parametrize $\mu_{tu}$ and $\sigma_t$ of a GGDM for all $t$ as sigmoid functions of trainable variables.
- **GGDM +PRED**: we parametrize all the $\mu_{tu}$ and $\sigma_t$ identically to GGDM, but also learn the marginal coefficients with a $\mathrm{cumsum} \circ \mathrm{softmax}$ parametrization (identical to VARS) instead of computing them via Theorem 1, as well as the coefficients that predict $\boldsymbol{x}_0$ from $a_t \boldsymbol{x}_t - b_t \boldsymbol{\epsilon}$ with $1 + \mathrm{softplus}$ and softplus parametrizations.
- **[family]+TIME**: for any sampler family, we additionally parametrize the timesteps used to query the score model with a $\mathrm{cumsum} \circ \mathrm{softmax}$ parametrization (identical to VARS).

As we will show in the experiments, despite the fact that our pre-trained DDPMs are trained with discrete timesteps, learning the timesteps is still helpful. In principle, this should only be possible for DDPMs trained with continuous time sampling (Chen et al., 2021a; Song et al., 2021b; Kingma et al., 2021), but in practice we find that DDPMs trained with continuously embedded discrete timesteps are still well-behaved when applied at timesteps not present during training. We think this is due to the regularity of the sinusoidal positional encodings Vaswani et al. (2017) used in these model architectures and training with a sufficiently large number of timesteps $T$.

### 3.1 Differentiable sample quality scores

We can differentiate through a stochastic sampler using the reparameterization trick, but the question of which objective to optimize still remains. Prior work has shown that optimizing log-likelihood can actually worsen sample quality and FID scores in the few-step regime (Watson et al., 2021; Song et al., 2021a). Thus, we instead design a *perceptual* loss which simply compares mean statistics between model samples and real samples in the neural network feature space. These types of objectives have been shown in prior work to better correlate with human perception of sample quality (Johnson et al., 2016; Heusel et al., 2017), which we also confirm in our experiments.

We rely on the representations of the penultimate layer of a pre-trained InceptionV3 classifier (Szegedy et al., 2016) and optimize the Kernel Inception Distance (KID) (Bińkowski et al., 2018). Let $\phi(\boldsymbol{x})$ denote the inception features of an image $\boldsymbol{x}$ and $p_\psi$ represent a diffusion sampler with trainable parameters $\psi$. For a linear kernel, which works best in our experiments, the objective is:

$$\mathcal{L}(\psi) = \mathop{\mathbb{E}}_{\boldsymbol{x}_p \sim p_\psi} \mathop{\mathbb{E}}_{\boldsymbol{x}_p' \sim p_\psi} \phi(\boldsymbol{x}_p)^\top \phi(\boldsymbol{x}_p') - \mathop{\mathbb{E}}_{\boldsymbol{x}_p \sim p_\psi} \mathop{\mathbb{E}}_{\boldsymbol{x}_q \sim q} \phi(\boldsymbol{x}_p)^\top \phi(\boldsymbol{x}_q) \qquad (7)$$

More generally, KID can be expressed as:

$$\mathcal{L}_{\text{KID}}(\psi) = \left\| \mathop{\mathbb{E}}_{\boldsymbol{x}_p \sim p_\psi} f^*(\boldsymbol{x}_p) - \mathop{\mathbb{E}}_{\boldsymbol{x}_q \sim q} f^*(\boldsymbol{x}_q) \right\|_2^2 , \qquad (8)$$

where $f^*(\boldsymbol{x}) = \mathbb{E}_{\boldsymbol{x}_p' \sim p_\psi} k_\phi(\boldsymbol{x}, \boldsymbol{x}_p') - \mathbb{E}_{\boldsymbol{x}_q' \sim q(\boldsymbol{x}_0)} k_\phi(\boldsymbol{x}, \boldsymbol{x}_q')$ is the witness function for any differentiable, positive definite kernel $k$, and $k_\phi(\boldsymbol{x}, \boldsymbol{y}) = k(\phi(\boldsymbol{x}), \phi(\boldsymbol{y}))$. Note that $f^*$ attains the supremum of the MMD. To enable stochastic gradient descent, we use an unbiased estimator of KID using a minibatch of $n$ model samples $\boldsymbol{x}_p^{(1)} \ldots \boldsymbol{x}_p^{(n)} \sim p_\psi$ and $n$ real samples $\boldsymbol{x}_q^{(1)} \ldots \boldsymbol{x}_q^{(n)} \sim q$:

$$\frac{1}{n(n-1)} \sum_{i \neq j}^{n} k_\phi(\boldsymbol{x}_p^{(i)}, \boldsymbol{x}_p^{(j)}) - \frac{2}{n^2} \sum_{i=1}^{n} \sum_{j=1}^{n} k_\phi(\boldsymbol{x}_p^{(i)}, \boldsymbol{x}_q^{(j)}) + c , \qquad (9)$$

where $c$ is constant in $\psi$. Since the sampling chain of any Gaussian diffusion process admits using the reparametrization trick, our loss function is fully differentiable with respect to the trainable variables $\psi$. We empirically find that using the perceptual features is crucial; i.e., by trying $\phi(\boldsymbol{x}) = \boldsymbol{x}$ to compare images directly on pixel space rather than neural network feature space (as above), we observe that our method makes samples consistently worsen in apparent quality as training progresses.

## 3.2 Gradient Rematerialization

In order for backpropagation to be feasible under reasonable memory constraints, one final problem must be addressed: since we are taking gradients with respect to model samples, the cost in memory to maintain the state of the forward pass scales linearly with the number of inference steps, which can quickly become unfeasible considering the large size of typical DDPM architectures. To address this issue, we use gradient rematerialization (Kumar et al., 2019b). Instead of storing a particular computation's output from the forward pass required by the backward pass computation, we recompute it on the fly. To trade $\mathcal{O}(K)$ memory cost for $\mathcal{O}(K)$ computation time, we simply rematerialize calls to the pre-trained DDPM (i.e., the estimated score function), but keep in memory all the progressively denoised images from the sampling chain. In JAX (Bradbury et al., 2018), this is trivial to implement by simply wrapping the score function calls with `jax.remat`.

## 4 Generalized Gaussian Diffusion Models

We now present Generalized Gaussian Diffusion Model (GGDM), our novel family of Gaussian diffusion processes that includes DDIM as a special case mentioned in section 3. We define a joint distribution with no independence assumptions

$$q_{\mu,\sigma}(\boldsymbol{x}_0, ..., \boldsymbol{x}_T) = q(\boldsymbol{x}_0)q(\boldsymbol{x}_T|\boldsymbol{x}_0)\prod_{t=1}^{T-1} q_{\mu,\sigma}(\boldsymbol{x}_t|\boldsymbol{x}_{>t}, \boldsymbol{x}_0) \tag{10}$$

where the new factors are defined as

$$q_{\mu,\sigma}(\boldsymbol{x}_t|\boldsymbol{x}_{>t}, \boldsymbol{x}_0) = \mathcal{N}\left(\boldsymbol{x}_t \left| \sum_{u \in S_t} \mu_{tu}\boldsymbol{x}_u, \sigma_t^2\boldsymbol{I}_d \right.\right) \tag{11}$$

(letting $S_t = \{0, ..., T\} \setminus \{1, ..., t\}$ for notation compactness), with $\sigma_t$ and $\mu_{tu}$ free parameters $\forall t \in \{1, ..., T\}, u \in S_t$. In other words, when predicting the next, less noisy image, the sampler can take into account *all* the previous, noisier images in the sampling chain, and similarly to DDIM, we can also control the sampler's variances. As we prove in the appendix (A.2), this construction admits Gaussian marginals, and we can differentiably compute the marginal coefficients from arbitrary choices of $\mu$ and $\sigma$:

**Theorem 1.** Given some $t \in \{1, ..., T\}$, let $a_{tu}^{(1)} = \mu_{tu} \ \forall u \in S_t$ and $v_t^{(1)} = \sigma_t^2$. For each $i \in \{1, ..., T - t\}$, recursively define

$$a_{tu}^{(i+1)} = a_{t,t+i}^{(i)}\mu_{t+i,u} + a_{tu}^{(i)} \ \forall u \in S_{t+i} \quad \text{and} \quad v_t^{(i+1)} = v_t^{(i)} + \left(a_{t,t+i}^{(i)}\sigma_{t+i}\right)^2.$$

Then, it follows that

$$q_{\mu,\sigma}(\boldsymbol{x}_t|\boldsymbol{x}_{>t+i}, \boldsymbol{x}_0) = \mathcal{N}\left(\boldsymbol{x}_t \left| \sum_{u \in S_{t+i}} a_{tu}^{(i+1)}\boldsymbol{x}_u, v_t^{(i+1)}\boldsymbol{I}_d \right.\right). \tag{12}$$

In other words, instead of letting the $\beta_t$ (or equivalently, the $\bar{\alpha}_t$) *define* the forward process as done by a usual DDPM, the GGDM family lets the $\mu_{tu}$ and $\sigma_t$ define the process. In particular, an immediate corollary of Theorem 1 is that the marginal coefficients are given by

$$q_{\mu,\sigma}(\boldsymbol{x}_t|\boldsymbol{x}_0) = \mathcal{N}\left(\boldsymbol{x}_t \left| a_{t0}^{(T-t+1)}\boldsymbol{x}_0, v_t^{(T-t+1)}\boldsymbol{I}_d \right.\right) \tag{13}$$

The reverse process is thus defined as $p(\boldsymbol{x}_T)\prod_{t=1}^{T} p(\boldsymbol{x}_{t-1}|\boldsymbol{x}_t)$ with $p(\boldsymbol{x}_T) \sim \mathcal{N}(\boldsymbol{0}, \boldsymbol{I}_d))$ and

$$p_\theta(\boldsymbol{x}_t|\boldsymbol{x}_{>t}) = q_{\mu,\sigma}\left(\boldsymbol{x}_t\middle|\boldsymbol{x}_{>t}, \hat{\boldsymbol{x}}_0 = \frac{1}{a_{t0}^{(T-t+1)}}\left(\boldsymbol{x}_t - \sqrt{v_t^{(T-t+1)}}\boldsymbol{\epsilon}_\theta(\boldsymbol{x}_t, t)\right)\right). \tag{14}$$

Table 1: FID and IS scores for DDSS against baseline methods for a DDPM trained on CIFAR10 with the $L_{\text{simple}}$ objective proposed by (Ho et al., 2020). FID scores (lower is better) are the numbers at the left of each entry, and IS scores (higher is better) are at the right.

| Sampler \ $K$ | 5 | 10 | 15 | 20 | 25 |
|---|---|---|---|---|---|
| DDPM (linear stride) | 84.27 / 5.396 | 43.39 / 7.034 | 31.40 / 7.609 | 25.94 / 7.879 | 22.60 / 8.043 |
| DDPM (quadratic stride) | 76.25 / 5.435 | 42.03 / 6.965 | 27.78 / 7.714 | 20.225 / 8.128 | 16.17 / 8.350 |
| DDIM (linear stride) | 44.41 / 6.750 | 19.11 / 7.965 | 14.06 / 8.190 | 11.82 / 8.420 | 10.52 / 8.512 |
| DDIM (quadratic stride) | 32.66 / 7.090 | 13.62 / 8.190 | 9.318 / 8.495 | 7.500 / 8.641 | 6.560 / 8.759 |
| GGDM +PRED+TIME | **13.77 / 8.520** | **8.227 / 8.903** | **6.115 / 9.050** | **4.722 / 9.261** | **4.250 / 9.186** |

Table 2: FID / IS scores for DDSS against baseline methods for a DDPM trained on ImageNet 64x64 with the $L_{\text{hybrid}}$ objective proposed by Nichol & Dhariwal (2021).

| Sampler \ $K$ | 5 | 10 | 15 | 20 | 25 |
|---|---|---|---|---|---|
| DDPM (linear stride) | 122.0 / 5.878 | 58.78 / 10.67 | 39.30 / 13.22 | 31.36 / 14.72 | 26.36 / 15.71 |
| DDPM (quadratic stride) | 394.8 / 1.351 | 129.5 / 5.997 | 80.10 / 9.595 | 61.34 / 11.60 | 49.60 / 13.01 |
| DDIM (linear stride) | 135.4 / 5.898 | 40.70 / 12.225 | 28.54 / 13.99 | 24.225 / 14.75 | 22.13 / 15.16 |
| DDIM (quadratic stride) | 409.1 / 1.380 | 148.6 / 5.533 | 67.65 / 9.842 | 45.60 / 11.99 | 36.11 / 13.225 |
| GGDM +PRED+TIME | **55.14 / 12.90** | **37.32 / 14.76** | **24.69 / 17.225** | **20.69 /17.92** | **18.40 / 18.12** |

## 4.1 Ignoring the Matching Marginals Condition

Unlike DDIM, the GGDM family does not guarantee that the marginals of the new forward process match that of the original DDPM. We empirically find, however, that this condition can often be too restrictive and better samplers exist where the marginals $q_{\mu,\sigma}(\boldsymbol{x}_t|\boldsymbol{x}_0) = \mathcal{N}\left(\boldsymbol{x}_t\middle|a_{t0}^{(T-t+1)}\boldsymbol{x}_0, v_t^{(T-t+1)}\boldsymbol{I}_d\right)$ of the new forward process differ from the original DDPM's marginals. We verify this empirically by applying DDSS to both the family of DDIM sigmas and DDPM variances ("VARS" in Section 3): both sampler families have the same number of parameters (the reverse process variances), but the latter does not adjust the mean coefficients like DDIM to ensure matching marginals and still achieves similar or better scores than the former across sample quality metrics (and even outperforms the DDIM($\eta = 0$) baseline); see Section 5.2.

## 5 Experiments

In order to emphasize that our method is compatible with any pre-trained DDPM, we apply our method on pre-trained DDPM checkpoints from prior work. Specifically, we experiment with the DDPM trained by Ho et al. (2020) with $L_{\text{simple}}$ on CIFAR10, as well as a DDPM following the exact configuration of Nichol & Dhariwal (2021) trained on ImageNet 64x64 (Deng et al., 2009) with their $L_{\text{hybrid}}$ objective (with the only difference being that we trained the latter ourselves for 3M rather than 1.5M steps). Both of these models utilize adaptations of the UNet architecture (Ronneberger et al., 2015) that incorporate self-attention layers (Vaswani et al., 2017).

We evaluate all of our models on both FID and Inception Score (IS) (Salimans et al., 2016), comparing the samplers discovered by DDSS against DDPM and DDIM baselines with linear and quadratic strides. As previously mentioned, more recent methods for fast sampling are outperformed by DDIM when the budget of inference steps is as small as those we utilize in this work (5, 10, 15, 20, 25). All reported results on both of these approximate sample quality metrics were computed by comparing 50K model and training data samples, as is standard in the literature. Also as is standard, IS scores are computed 10 times, each time on 5K samples, and then averaged.

In all of our experiments, we optimize the DDSS objective presented in Section 3.1 with the following hyperparameters:

1. For every family of models we search over, we initialize the degrees of freedom such that training begins with a sampler matching DDPM with $K$ substeps following Song et al. (2020); Nichol & Dhariwal (2021).

Table 3: FID / IS scores for the KID kernel ablation on CIFAR10. When not learning the timesteps, we fix them to a quadratic stride, as Table 1 shows this performs best on CIFAR10.

| Sampler \ $K$ | 5 | 10 | 15 | 20 | 25 |
|---|---|---|---|---|---|
| DDSS (linear kernel) | | | | | |
|    GGDM +PRED+TIME | 13.77 / 8.520 | 8.227 / 8.903 | 6.115 / 9.050 | **4.722** / **9.261** | **4.250** / **9.186** |
|    GGDM +PRED | 14.26 / 8.406 | 8.617 / 8.842 | 5.939 / 9.035 | 4.893 / 9.153 | 4.574 / 9.145 |
|    GGDM +TIME | 12.85 / 8.383 | **7.858** / 8.895 | 6.265 / **9.075** | 5.367 / 9.136 | 4.887 / 9.229 |
|    GGDM) | 14.45 / 8.281 | 8.154 / 8.892 | 7.045 / 8.939 | 5.477 / 9.183 | 4.815 / 9.189 |
| DDSS (cubic kernel) | | | | | |
|    GGDM +PRED+TIME | 14.41 / **8.527** | 8.2257 / **9.007** | **5.895** / 9.036 | 4.932 / 9.092 | 4.278/ **9.286** |
|    GGDM +PRED | 14.39 / 8.401 | 8.977 / 8.870 | 6.517 / 8.970 | 4.915 / 9.132 | 4.471 / 9.247 |
|    GGDM +TIME | **12.35** / 8.406 | 7.879 / 8.852 | 6.682 / 8.999 | 5.639 / 9.058 | 4.631 / 9.189 |
|    GGDM | 14.57 / 8.297 | 8.2252 / 8.836 | 6.727 / 8.904 | 5.569 / 9.177 | 4.668 / 9.192 |

2. We apply gradient updates using the Adam optimizer (Kingma & Ba, 2015). We swept over the learning rate and used $\lambda = 0.0005$. We did not sweep over other Adam hyperparameters and kept $\beta_1 = 0.9$, $\beta_2 = 0.999$, and $\epsilon = 1 \times 10^{-8}$.
3. We tried batch sizes of 128 and 512 and opted for the latter, finding that it leads to better sample quality upon inspection. Since the loss depends on averages over examples as our experiments are on unconditional generation, this choice was expected.
4. We run all of our experiments for 50K training steps and evaluate the discovered samplers at this exact number of training steps. We did not sweep over this value.

We include our main results in Table 1 for CIFAR10 and Table 2 for ImageNet 64x64, comparing DDSS applied to GGDM +PRED+TIME against DDPM and DDIM baselines with linear and quadratic strides. All models use a linear kernel, i.e., $k_\phi(\boldsymbol{x}, \boldsymbol{y}) = \phi(\boldsymbol{x})^\top \phi(\boldsymbol{y})$, which we found to perform slightly better than the cubic kernel used by Bińkowski et al. (2018) (we ablate this in section 5.1). We omit the use of the learned variances of the ImageNet 64x64 model (i.e., following Nichol & Dhariwal (2021)), as we search for the variances ourselves via DDSS. We include samples for 5, 10 and 25 steps comparing the strongest DDIM baselines to DDSS + GGDM with a learned stride; see Figures 1 and 3. We include additional ImageNet 64x64 samples (A.1) and results for larger resolution datasets (A.4) in the appendix.

## 5.1 ABLATIONS FOR KID KERNEL AND GGDM VARIANTS

As our approach is compatible with any choice of KID kernel, we experiment with different choices of kernels. Namely, we try the simplest possible linear kernel, $k_\phi(\boldsymbol{x}, \boldsymbol{y}) = \phi(\boldsymbol{x})^\top \phi(\boldsymbol{y})$, as well as the cubic kernel $k_\phi(\boldsymbol{x}, \boldsymbol{y}) = \left(\frac{1}{d}\phi(\boldsymbol{x})^\top \phi(\boldsymbol{y}) + 1\right)^3$ used by Bińkowski et al. (2018). We compare the performance of these two kernels, as well as different variations of GGDM (i.e., with and without TIME and PRED as defined in Section 3). Results are included for CIFAR10 across all budgets in Table 3. We also include a smaller version of this ablation for ImageNet 64x64 in the appendix (A.3).

The results in this ablation show that the contributions of the linear kernel, timestep learning, and the empirical choice of learning the coefficients that predict $\boldsymbol{x}_0$ all slightly contribute to better FID and IS scores. Importantly, however, removing any of these additions still allows us to comfortably outperform the strongest baselines. See also the results on LSUN in the appendix A.4, which also do not include these additional trainable variables.

## 5.2 SEARCH SPACE ABLATION

Now, in order to further demonstrate the key importance of optimizing our GGDM family to find high-fidelity samplers, we also apply DDSS to the less general DDIM and VARS families. We show that, while we still attain better scores than a regular DDPM, searching these less flexible families of samplers does not yield improvements as significant as with out novel GGDM family. In particular, optimizing the DDIM sigma coefficients does not outperform the corresponding DDIM($\eta = 0$) baseline on CIFAR10, which is not a surprising result as Song et al. (2020) show empirically that most choices of the $\sigma_t$ degrees of freedom lead to worse FID scores than setting them all to 0. These

Table 4: FID / IS scores for the DDSS search space ablation on CIFAR10. All runs fix the timesteps to a quadratic stride and use a linear kernel except for the last row (we only include the GGDM results for ease of comparison).

| Sampler \ $K$ | 5 | 10 | 15 | 20 | 25 |
|---|---|---|---|---|---|
| DDIM($\eta = 0$) | 32.66 / 7.090 | 13.62 / 8.190 | 9.318 / 8.495 | 7.500 / 8.641 | 6.560 / 8.759 |
| DDSS | | | | | |
|    VARS | 33.08 / 7.096 | 15.33 / 8.559 | 9.693 / 8.845 | 7.297 / 8.924 | 6.172 / 9.057 |
|    DDIM | 32.61 / 7.084 | 16.29 / 7.966 | 11.31 / 8.372 | 9.120 / 8.563 | 7.853 / 8.644 |
|    GGDM | 14.45 / 8.281 | 8.154 / 8.892 | 7.045 / 8.939 | 5.477 / 9.183 | 4.815 / 9.189 |
|    GGDM +PRED+TIME | 13.77 / 8.520 | 8.227 / 8.903 | 6.115 / 9.050 | 4.722 / 9.261 | 4.250 / 9.186 |

results also show that optimizing the VARS can outperform DDSS applied to the DDIM family, and even the strongest DDIM($\eta = 0$) baselines for certain budgets, justifying our choice of not enforcing the marginals to match (as discussed in Section 4.1).

# 6  DISCUSSION

When applied to a sufficiently flexible family (such as the GGDM family proposed in this work), DDSS consistently finds samplers that achieve better image generation quality than the strongest baselines in the literature for very few steps. This is qualitatively apparent in non-cherrypicked samples (*e.g.,* DDIM($\eta = 0$) tends to generate blurrier images and with less background details as the budget decreases), and multiple quantitative sample quality metrics (FID and IS) also reflect these results. Still, we observe limitations to our method. Finding samplers with inference step budgets as small as $K < 10$ that have little apparent loss in quality remains challenging with our proposed search family. And, while on CIFAR10 the metrics indicate significant relative improvement over sample quality metrics, the relative improvement on ImageNet 64x64 is less pronounced. We hypothesize that this is an inherent difficulty of ImageNet due to its high diversity of samples, and that in order to retain sample quality and diversity, it might be impossible to escape some minimum number of inference steps with score-based models as they might be crucial to mode-breaking.

Beyond the empirical gains of applying our procedure, our findings shed further light into properties of pre-trained score-based generative models. First, we show that without fine-tuning a DDPM's parameters, these models are already capable of producing high-quality samples with very few inference steps, though the default DDPM sampler in this regime is usually suboptimal when using a few-step sampler. We further show that better sampling paths exist, and interestingly, these are determined by alternative variational lower bounds to the data distribution that make use of the score-based model but do not necessarily share the same marginals as the original DDPM forward process. Our findings thus suggest that enforcing this marginal-sharing constraint is unnecessary and can be too restrictive in practice.

# 7  OTHER RELATED WORK

Besides DDIM (Song et al., 2020), there have been more recent attempts at reducing the number of inference steps for DDPMs. Jolicoeur-Martineau et al. (2021) proposed a dynamic step size SDE solver that can reduce the number of calls to the modeled score function to $\sim 150$ on CIFAR10 (Krizhevsky et al., 2009) with minimal cost in FID scores, but quickly falls behind DDIM($\eta = 0$) with as many as 50 steps. Watson et al. (2021) proposed a dynamic programming algorithm that chooses log-likelihood optimal strides, but find that log-likelihood reduction has a mismatch with FID scores, particularly with in the very few step regime, also falling behind DDIM($\eta = 0$) in this front. Other methods that have been shown to help sample quality in the few-step regime include non-Gaussian variants of diffusion models (Nachmani et al., 2021) and adaptively adjusting the sampling path by introducing a noise level estimating network (San-Roman et al., 2021), but more thorough evaluation of sample quality achieved by these approaches is needed with budgets as small as those considered in this work.

Other approaches to sampling DDPMs have also been recently proposed, though not for the explicit purpose of efficient sampling. Song et al. (2021b) derive a reverse SDE that, when discretized, uses

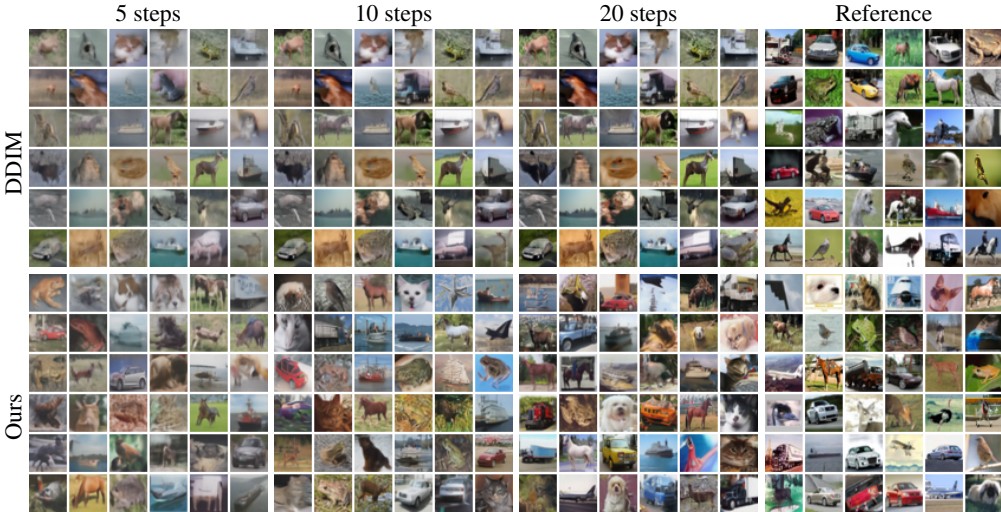

Figure 3: Non-cherrypicked samples for a DDPM trained on CIFAR10, comparing the strongest DDIM($\eta = 0$) baseline and our approach. All samples were generated with the same random seeds. For reference, we include DDPM samples using all 1,000 steps (top right) and real images (bottom right).

different coefficients than the ancestral samplers considered in this work. The same authors also derive "corrector" steps, which introduce additional calls to the pre-trained DDPM as a form of gradient ascent (Langevin dynamics) that help with quality but introduce computation cost, as well as an alternative sampling procedure using a probability flow ODE that shares the same marginals as the DDPM's original forward process. Huang et al. (2021) generalize this family of samplers to a "plug-in reverse SDE" that interpolates between a probability flow ODE and the reverse SDE, similarly to how the DDIM $\eta$ interpolates between an implicit probabilistic model and a stochastic reverse process. Our proposed search family includes discretizations of most of these cases for Gaussian processes, notably missing corrector steps, where reusing a single timestep is considered.

## 8    CONCLUSION AND FUTURE WORK

We propose Differentiable Diffusion Sampler Search (DDSS), a method for finding few-step samplers for Denoising Diffusion Probabilistic Models. We show how to optimize a perceptual loss over a space of diffusion processes that makes use of a pre-trained DDPM's samples by leveraging the reparametrization trick and gradient rematerialization. Our results qualitatively and quantitatively show that DDSS is able to significantly improve sample quality for unconditional image generation over prior methods on efficient DDPM sampling. The success of our method hinges on searching a novel, wider family of Generalized Gaussian Diffusion Model (GGDM) than those identified in prior work (Song et al., 2020). DDSS does not fine-tune the pre-trained DDPM, only needs to be applied once, has few hyperparameters, and does not require re-training the DDPM.

Our findings suggest future directions to further reduce the number of inference steps while retaining high fidelity in generated samples. For instance, it is plausible to use different representations for the perceptual loss instead of those of a classifier, *e.g.,* use representations from an unsupervised model such as SimCLR (Chen et al., 2020), to using internal representations learned by the pre-trained DDPM itself, which would eliminate the burden of additional computation. Moreover, considering the demonstrated benefits of applying DDSS to our proposed GGDM family of samplers (as opposed to narrower families like DDIM), we motivate future work on identifying more general families of samplers and investigating whether they help uncover even better samplers or lead to overfitting. Finally, identifying other variants of perceptual losses (*e.g.,* that do not sample from the model), or alternative optimization strategies (*e.g.,* gradient-free methods) that lead to similar results is important future work. This would make DDSS itself a more efficient procedure, as gradient-based optimization of our proposed loss requires extensive memory or computation requirements to back-propagate through the whole sampling chain.

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

# A APPENDIX

## A.1 ADDITIONAL IMAGENET 64x64 SAMPLES

We provide additional samples for our results on ImageNet 64x64. The DDPM and DDIM($\eta = 0$) samples (left and middle, respectively) use a linear stride, while our DDSS + GGDM samples (right) use a learned stride.

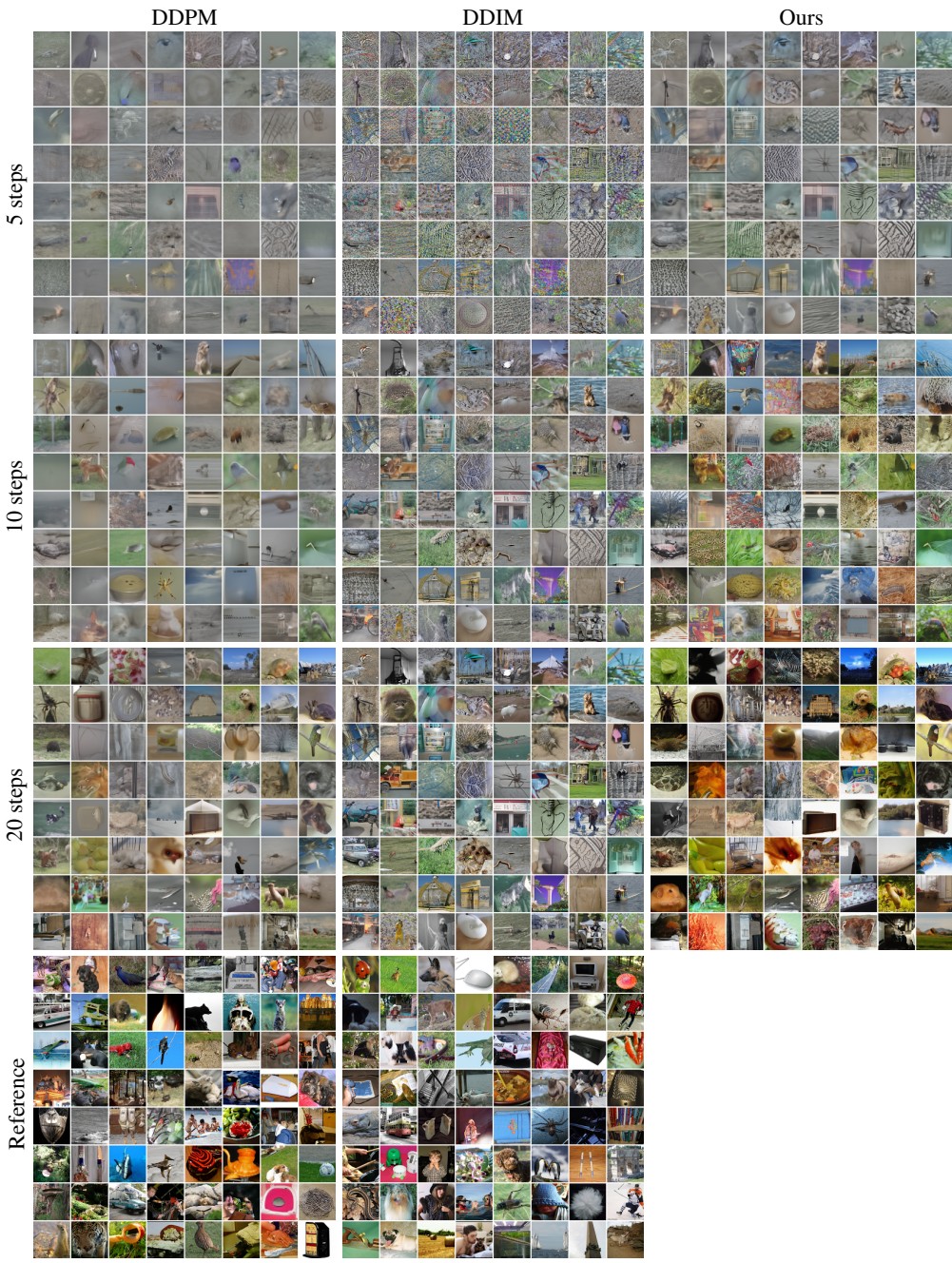

Figure A.1: Additional samples on ImageNet 64x64. For reference, we include DDPM samples with all 4,000 steps (bottom left) and real samples (bottom middle).

## A.2 PROOF FOR THEOREM 1

**Theorem 1.** Given some $t \in \{1, ..., T\}$, let $a_{tu}^{(1)} = \mu_{tu} \; \forall u \in S_t$ and $v_t^{(1)} = \sigma_t^2$. For each $i \in \{1, ..., T - t\}$, recursively define

- $a_{tu}^{(i+1)} = a_{t,t+i}^{(i)} \mu_{t+i,u} + a_{tu}^{(i)} \; \forall u \in S_{t+i}$

- $v_t^{(i+1)} = v_t^{(i)} + \left( a_{t,t+i}^{(i)} \sigma_{t+i} \right)^2$

Then, it follows that

$$q_{\mu,\sigma}(\boldsymbol{x}_t | \boldsymbol{x}_{>t+i}, \boldsymbol{x}_0) = \mathcal{N} \left( \boldsymbol{x}_t \, \middle| \, \sum_{u \in S_{t+i}} a_{tu}^{(i+1)} \boldsymbol{x}_u, v_t^{(i+1)} \boldsymbol{I}_d \right)$$

*Proof.* Let us prove this result with mathematical induction. Note that, for each such $t$, we have by definition that

$$q_{\mu,\sigma}(\boldsymbol{x}_{t+1} | \boldsymbol{x}_{>t+1}, \boldsymbol{x}_0) = \mathcal{N} \left( \boldsymbol{x}_t \, \middle| \, \sum_{u \in S_{t+1}} \mu_{t+1,u} \boldsymbol{x}_u, \sigma_{t+1}^2 \boldsymbol{I}_d \right)$$

and

$$q_{\mu,\sigma}(\boldsymbol{x}_t | \boldsymbol{x}_{t+1}, \boldsymbol{x}_{>t+1}, \boldsymbol{x}_0) = \mathcal{N} \left( \boldsymbol{x}_t \, \middle| \, \sum_{u \in S_t} \mu_{tu} \boldsymbol{x}_u, \sigma_t^2 \boldsymbol{I}_d \right) = \mathcal{N} \left( \boldsymbol{x}_t \, \middle| \, \sum_{u \in S_t} a_{tu}^{(1)} \boldsymbol{x}_u, v_t^{(1)} \boldsymbol{I}_d \right)$$

Therefore, following Svensén & Bishop (2007) (2.115), by prior conjugacy it follows that

$$
\begin{aligned}
q_{\mu,\sigma}(\boldsymbol{x}_t | \boldsymbol{x}_{>t+1}, \boldsymbol{x}_0) &= \mathcal{N} \left( \boldsymbol{x}_t \, \middle| \, a_{t,t+1}^{(1)} \sum_{u \in S_{t+1}} \mu_{t+1,u} \boldsymbol{x}_u + \sum_{u \in S_{t+1}} a_{tu}^{(1)}, \left( v_t^{(1)} + a_{t,t+1}^{(1)} \sigma_{t+1}^2 a_{t,t+1}^{(1)} \right) \boldsymbol{I}_d \right) \\
&= \mathcal{N} \left( \boldsymbol{x}_t \, \middle| \, \sum_{u \in S_{t+1}} \left( a_{t,t+1}^{(1)} \mu_{t+1,u} + a_{tu}^{(1)} \right) \boldsymbol{x}_u, v_t^{(2)} \boldsymbol{I}_d \right) \\
&= \mathcal{N} \left( \boldsymbol{x}_t \, \middle| \, \sum_{u \in S_{t+1}} a_{tu}^{(2)} \boldsymbol{x}_u, v_t^{(2)} \boldsymbol{I}_d \right).
\end{aligned}
$$

This proves the base case for our induction argument. Now, let us prove the inductive step. Suppose there exists some integer $j \in \{1, ..., T - t + 1\}$ such that

$$q_{\mu,\sigma}(\boldsymbol{x}_t | \boldsymbol{x}_{>t+j}, \boldsymbol{x}_0) = \mathcal{N} \left( \boldsymbol{x}_t \, \middle| \, \sum_{u \in S_{t+j}} a_{tu}^{(j+1)} \boldsymbol{x}_u, v_t^{(j+1)} \boldsymbol{I}_d \right).$$

By definition, we already know $q(\boldsymbol{x}_{t+j+1} | \boldsymbol{x}_{>t+j+1}, \boldsymbol{x}_0)$, so we have

$$q_{\mu,\sigma}(\boldsymbol{x}_{t+j+1} | \boldsymbol{x}_{>t+j+1}, \boldsymbol{x}_0) = \mathcal{N} \left( \boldsymbol{x}_{t+j+1} \, \middle| \, \sum_{u \in S_{t+j+1}} \mu_{t+j+1,u} \boldsymbol{x}_u, \sigma_{t+j+1}^2 \boldsymbol{I}_d \right)$$

and (rewriting the inductive hypothesis)

$$q_{\mu,\sigma}(\boldsymbol{x}_t | \boldsymbol{x}_{t+j+1}, \boldsymbol{x}_{>t+j+1}, \boldsymbol{x}_0) = \mathcal{N} \left( \boldsymbol{x}_t \, \middle| \, \sum_{u \in S_{t+j}} a_{tu}^{(j+1)} \boldsymbol{x}_u, v_t^{(j+1)} \boldsymbol{I}_d \right).$$

Therefore, by prior conjugacy again, it follows that

$$
\begin{aligned}
&q_{\mu,\sigma}(\boldsymbol{x}_t | \boldsymbol{x}_{>t+j+1}, \boldsymbol{x}_0) \\
=&\mathcal{N} \left( \boldsymbol{x}_t \, \middle| \, a_{t,t+j+1}^{(j+1)} \sum_{u \in S_{t+j}} \mu_{t+j+1,u} \boldsymbol{x}_u + \sum_{u \in S_{t+j}} a_{tu}^{(j+1)}, \left( v_t^{(j+1)} + a_{t,t+j+1}^{(j+1)} \sigma_{t+j+1}^2 a_{t,t+j+1}^{(j+1)} \right) \boldsymbol{I}_d \right) \\
=&\mathcal{N} \left( \boldsymbol{x}_t \, \middle| \, \sum_{u \in S_{t+j}} \left( a_{t,t+j+1}^{(j+1)} \mu_{t+j+1,u} + a_{tu}^{(j+1)} \right) \boldsymbol{x}_u, v_t^{(j+2)} \boldsymbol{I}_d \right) \\
=&\mathcal{N} \left( \boldsymbol{x}_t \, \middle| \, \sum_{u \in S_{t+j+1}} a_{tu}^{(j+2)} \boldsymbol{x}_u, v_t^{(j+2)} \boldsymbol{I}_d \right).
\end{aligned}
$$

This concludes the proof of the inductive step. Hence, we have proven the result for any $i \in \{1, ..., T - t\}$. In particular,

$$q_{\mu,\sigma}(\boldsymbol{x}_t | \boldsymbol{x}_0) = \mathcal{N} \left( \boldsymbol{x}_t \, \middle| \, a_{t0}^{(T-t+1)} \boldsymbol{x}_0, v_t^{(T-t+1)} \boldsymbol{I}_d \right). \quad \square$$

### A.3 Additional ablation of KID kernel and GGDM variants for ImageNet 64x64

We also ran a smaller version of the ablation results presented in Section 5.1, but for ImageNet 64x64 instead of CIFAR10, as these are more computationally intensive to do a full grid search. Results for a step budget $K = 15$ are included below. When not learning the timesteps, we fix them to a linear stride, as Table 2 shows this performs best on ImageNet 64x64.

| Sampler $\setminus K$ | 15 |
|---|---|
| DDSS (linear kernel) | |
|    GGDM +PRED+TIME | **24.69** / 17.225 |
|    GGDM +PRED | 27.08 / 16.44 |
|    GGDM +TIME | 25.73 / **17.27** |
|    GGDM | 28.34 / 16.63 |
| DDSS (cubic kernel) | |
|    GGDM +PRED+TIME | 26.52 / 16.29 |
|    GGDM +PRED | 27.82 / 16.3 |
|    GGDM +TIME | 26.87 / 16.99 |
|    GGDM | 28.83 / 16.32 |

### A.4 Results on larger resolution datasets

We include results for LSUN (Yu et al., 2015) bedrooms and churches at the 128x128 resolution. We trained the models for 400K and 200K steps (respectively), and all other hyperparameters are identical: we use the Adam optimizer with learning rate 0.0003 (linearly warmed up for the first 1000 training steps), batch size 2048, gradient clipping at norms over 1.0, dropout of 0.1, and EMA over the weights with decay rate 0.9999. We train the models using a linear stride of 1000 evenly-spaced timesteps, fixing the log-signal-to-noise-ratio schedule to a cosine function monotonically decreasing from 20 to -20. The ELBO is reweighted with $L_{\text{simple}}$ following Ho et al. (2020), but we additionally reweight each term by $\max(1, \text{SNR})$ which we found to be slightly helpful in resulting FID scores (note this is equivalent to minimizing the worst mean squared error between either the $x_0$ or $\epsilon$). The UNet employs five down/up-sampling resolutions with $768 \times (1, 2, 4, 6, 8)$ respective channels, 3 ResNet blocks per resolution, and spatial self-attention at the 3 smallest resolutions, i.e., 8, 16, and 32.

After training the models, we run DDSS using just the GGDM model family for simplicity (i.e., we don't use the +PRED and +TIME we experiment with in the paper) at 5, 10 and 20 evenly-spaced inference steps. DDSS training occurs for 50K steps, using the Adam optimizer with learning rate of 0.0005 and batch size 512, optimizing the linear kernel for the KID loss. We compare against the usual DDPM and DDIM($\eta = 0$) baselines at the same inference budgets and include the FID scores with all 1000 steps for reference. Results and samples are included below.

| Sampler $\setminus K$ | 5 | 10 | 20 | 1000 |
|---|---|---|---|---|
| LSUN Bedroom | | | | |
|    DDPM | 95.38 | 44.84 | 16.88 | **2.547** |
|    DDIM($\eta = 0$) | 168.7 | 56.33 | 9.527 | – |
|    DDSS (GGDM) | **29.15** | **11.01** | **4.817** | – |
| LSUN Church | | | | |
|    DDPM | 96.67 | 51.05 | 16.53 | **2.718** |
|    DDIM($\eta = 0$) | 133.1 | 54.39 | 14.96 | – |
|    DDSS (GGDM) | **30.24** | **11.59** | **6.736** | – |

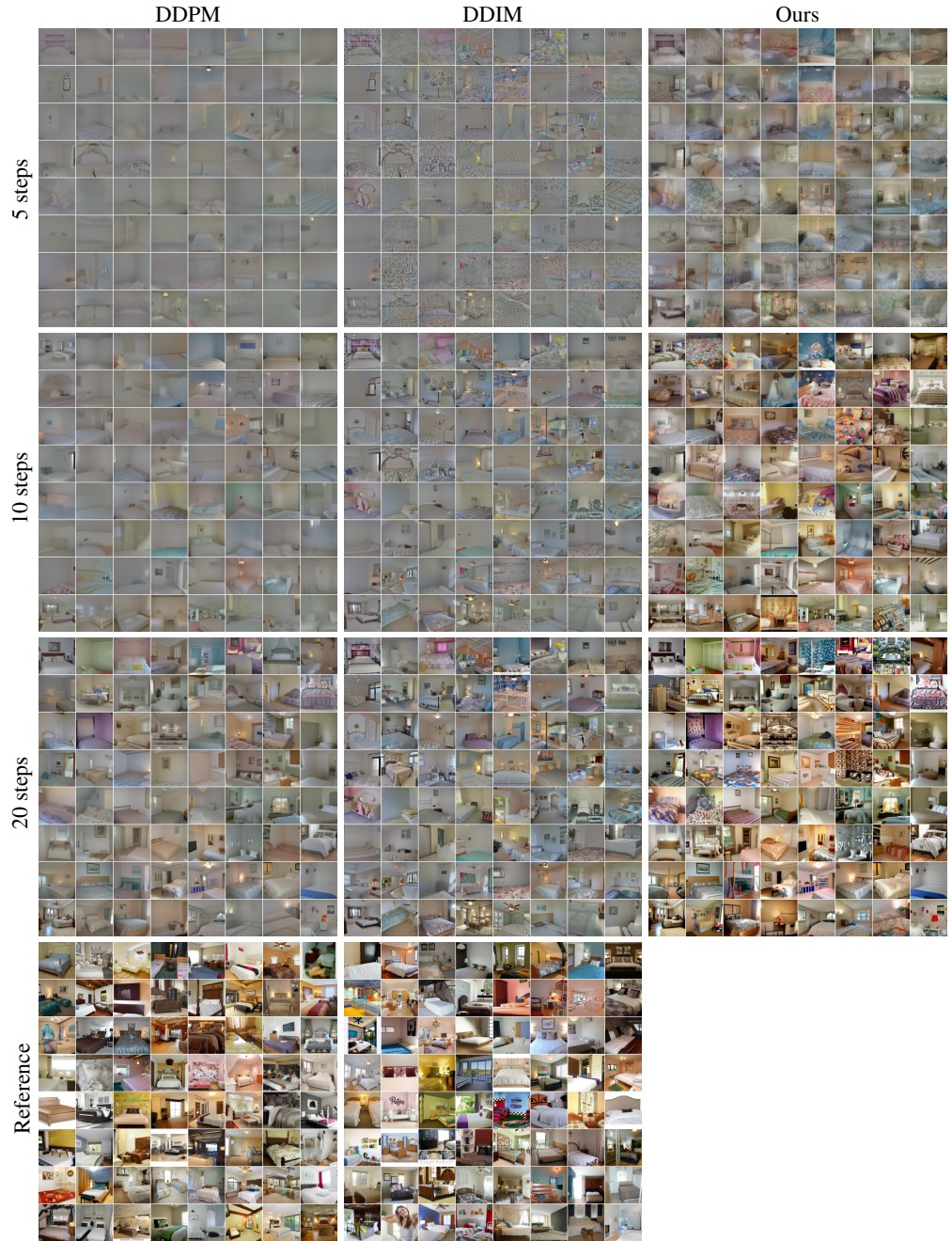

Figure A.2: Non-cherrypicked samples for a DDPM trained on LSUN bedroom 128x128, comparing DDPM and DDIM($\eta = 0$) to our approach. All samples were generated with the same random seeds and a linear stride. For reference, we include DDPM samples using all 1,000 steps (bottom left) and real images (bottom middle).

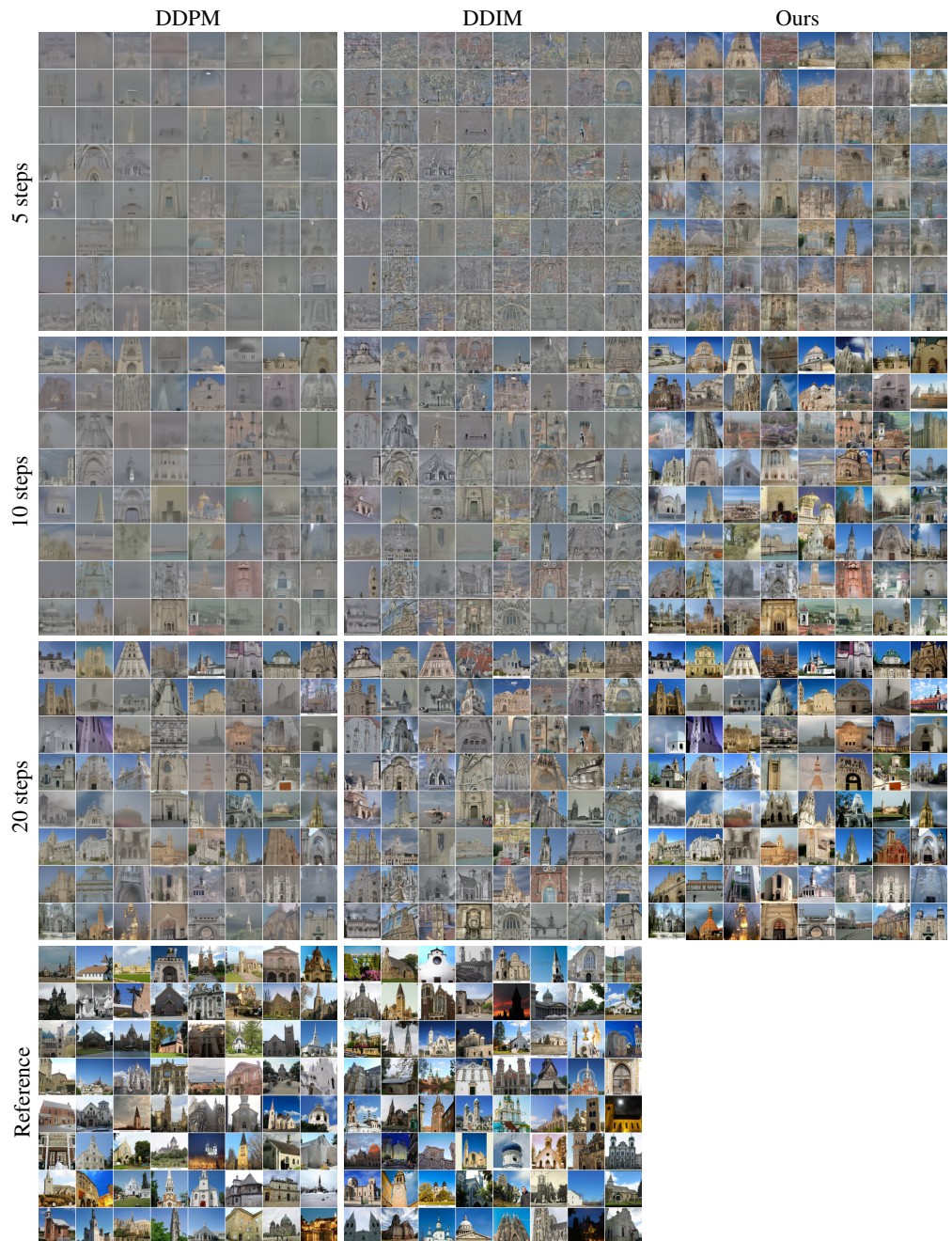

Figure A.3: Non-cherrypicked samples for a DDPM trained on LSUN church 128x128, comparing DDPM and DDIM($\eta = 0$) to our approach. All samples were generated with the same random seeds and a linear stride. For reference, we include DDPM samples using all 1,000 steps (bottom left) and real images (bottom middle).

