# OpenReview forum: "Learning Fast Samplers for Diffusion Models by Differentiating Through Sample Quality"
_ICLR.cc/2022/Conference — ICLR 2022 Poster_

### Official Review · Reviewer_vLgy · 2021-10-29

**Correctness:** 4
**Technical Novelty And Significance:** 3
**Empirical Novelty And Significance:** 4
**Recommendation:** 6
**Confidence:** 4

**Main Review:**

**Strengths**

The idea in the paper is relatively simple and straightforward, and has good empirical performance. The paper discusses how to address the engineering challenges in general DDSS search problems (discontinuity of timesteps, memory constraints). Empirically, optimizing perceptual losses benefits sample quality and is better than evaluation likelihoods (especially when most methods need many steps to have marginally better likelihoods).

**Weaknesses**

The clarity of the paper is a bit on the weaker side, and I think the paper can be significantly improved by having better clarity.
- It was quite unclear at first glance what the double subscripts mean in Equation (2) and beyond. It should be made explicit that subscript $ts$ means starting from $t$ and ending at $s$. These double subscripts also make it difficult to identify the main idea behind GGDP. Perhaps having a figure illustrating the GGDP idea helps with clarity.
- It is unclear whether, with the changed GGDP variational distribution, it is still possible to recover the same "diffusion / score-matching" solution with a new variational objective (not defined in the paper). I am inclined to believe this is the case, but I think a proof (like the one in DDIM paper) would be warranted.
- Equation (9) should be written explicitly as the batch version, since the actual algorithm is not an unbiased estimator of (9). Also is there a reason behind only matching the first-order moment, rather than higher-order moments? Is there a possibility that the DDSS sampler collapses to a single point that has the correct mean? This might need a bit of explanation.
- Minor, it would be helpful to have more details on how the parametrization in DDSS is performed in the appendix. Part of the confusion may be from the fact that we have a lot of notations, such as $\mu_{tu}$ and $\sigma_t$, $a$, $v$ etc. It would be great to explicitly explain what these mean (or correspond to in DDPM / DDIM).
- I have a hard time understanding the code `t = cumsum(softmax([*v, 1.]))[:-1]`. What are the range of v and t? Maybe an example in the appendix can help?
- I think the proof sketch can be moved into appendix as a more formal proof. At a high-level everything is Gaussian, so the claims themselves are not too hard to understand.

**Comments**

- The GGDP methods seem to be quite related to multistep methods in numerical integration, [here](https://en.wikipedia.org/wiki/Linear_multistep_method). I wonder what would be the connections to established multistep methods, such as Adam-Bashforth? It would be interesting to see what parameters does DDSS learn to further understand what it does.
- Is there a reason why datasets with larger images such as LSUN (256x256) are not tested?


==== Post rebuttal review ===
I think the rebuttal addressed most of my concerns. I am a bit disappointed that no discussion about larger image datasets are given, but the other points are adequately addressed. So I will keep my score as is.

**Summary Of The Paper:**

The paper discusses how to accelerate a diffusion generative model via Differentiable Diffusion Sampler Search (DDSS). First, a family of non-Markovian samplers is introduced, allowing the models to use multiple previous steps for generation. Next, a sampler search method is proposed to optimize the perceptual loss, with certain engineering issues addressed such as bounded and monotonically increasing steps and gradient parametrization. Empirically, this is able to improve sample quality in a few steps compared to other heuristics such as linear and quadratic schedules on CIFAR10 and ImageNet 64x64, especially when there is a very limited amount of steps that can be taken. The proposed DDSS + GGDP method outperforms DDIM and DDSS + DDIM, showing the advantage of DDSS.

**Summary Of The Review:**

I think the problem of accelerating diffusion models is important and believe that the paper makes a novel contribution towards it. That being said, I believe that the paper has serious issues regarding clarity and minor issues on experiments over larger datasets.

I am willing to increase the score if the authors address clarity issues well enough in the rebuttal and especially in the revisions that are allowed by the ICLR format. If the revision does not change by much, then my score will be around 5 and 6 (good idea, relatively poorly polished paper).

---

> ### Author Response · Authors · 2021-11-23
> **Reply to reviewer vLgy**
>
> Thank you for your valuable review and feedback. We address some of your methodology concerns below.
>
> > It is unclear whether, with the changed GGDP variational distribution, it is still possible to recover the same "diffusion / score-matching" solution with a new variational objective (not defined in the paper). I am inclined to believe this is the case, but I think a proof (like the one in DDIM paper) would be warranted.
>
> This solution is indeed recovered. We have now explicitly included the constructions of the reverse process and the ELBO in Section 3, in particular, showing how the diffusion KL terms can be computed analytically (similarly to DDPMs) as a mean squared error between the true and predicted $\epsilon$, which corresponds to denoising score matching.
>
> > The GGDP methods seem to be quite related to multistep methods in numerical integration, here. I wonder what would be the connections to established multistep methods, such as Adam-Bashforth? It would be interesting to see what parameters does DDSS learn to further understand what it does.
>
> Thank you for pointing out this connection that we weren’t aware of. We will investigate this area further.
>
> > Equation (9) should be written explicitly as the batch version, since the actual algorithm is not an unbiased estimator of (9). Also is there a reason behind only matching the first-order moment, rather than higher-order moments? Is there a possibility that the DDSS sampler collapses to a single point that has the correct mean? This might need a bit of explanation.
>
> Please see our general reply to all reviewers and the updated paper, where we now optimize for the Kernel Inception Distance (KID). With this loss, we now minimize an *unbiased* estimator of the squared maximum mean discrepancy (MMD) between the data and model distributions. Moreover, we also now include results for a cubic kernel in the appendix, which attains similar (albeit slightly worse) results than the linear kernel and does not admit this degenerate solution.
>
> We agree with all the feedback you have provided in regards to the clarity of the paper and have incorporated the changes into the revised version of the paper. Thank you!

---

### Official Review · Reviewer_u4ae · 2021-11-02

**Correctness:** 3
**Technical Novelty And Significance:** 2
**Empirical Novelty And Significance:** 2
**Recommendation:** 6
**Confidence:** 4

**Main Review:**

1. It seems that optimizing the perceptual loss might not give you the samples from the true underlying DDPM model. Although this approach might produce reasonable samples empirically, it lacks theoretical insights: it is unclear what distribution these samples are optimized to match with. It seems that the training objective is hand-crafted, and might lack theoretical guarantees.
2. Although samples from the proposed method have good FID/IS scores, optimizing the model via the perceptual loss could be directly/indirectly overfitting the evaluation metrics. So the improvements in FID/IS scores might be expected.
3. Are you still able to get likelihoods (lower bound) on the generated samples from GGDP?
4. As mentioned in the paper, GGDP can use pre-trained DDPM models. It would be more convincing to also show results on more challenging high-resolution datasets (e.g. LSUN, ImageNet, CelebA) using pre-trained models since the inference time for high-resolution datasets are often significantly longer, and improving the inference time would be a more crucial task for larger models. Currently, only small-scale datasets are considered.
5. For theorem 1, there is only "proof sketch". Is there a formal proof?
6. In section 4.1, it would be better to mention clearly what "the collection of all the introduced trainable variables" is.
7. It seems that the samples of the proposed method in figure 2 look reasonable when using 5 steps on CIFAR-10. However, in the appendix, the samples using 5 steps on ImageNet 64 do not look that realistic. Does it imply that the performance of the proposed approach is dependent on the dataset? If that is the case, more datasets are needed to show the strength of the method.

**Summary Of The Paper:**

Denoising Diffusion Probabilistic Models (DDPMs) can generate high-quality samples, but they are not efficient at inference. This paper proposes Differentiable Diffusion Sampler Search (DDSS) to generate high-quality samples while using fewer inference steps than DDPM. The proposed approach uses reparametrization trick and gradient rematerialization to optimize over a class of parametric samplers that use fewer steps than DDPMs. By optimizing perceptual losses, the proposed approach can generate high-quality samples using a smaller number of inference steps compared to existing approaches for sampling from DDPMs.

**Summary Of The Review:**

1. Although the proposed approach can generate high-quality samples, the objective used is hand-crafted and might lack theoretical guarantees. As mentioned in the paper "GGDP family does not guarantee that the marginals of the new forward process match that of the original DDPM." Although the sample quality can be reasonable, it is unclear what the model is doing theoretically (e.g., is it optimizing likelihood, or doing score matching).
2. The samples, after optimizing perceptual loss, might not correspond to the samples of the underlying DDPM model. Optimizing via perceptual distance is expected to have better performance under FID/IS scores, but it would also change the statistical property of DDPM. In some sense, the samples are not from DDPM, and it might be useful to discuss in more detail the statistical properties of GGDP.
3. As mentioned in the paper, GGDP can use pre-trained DDPM models. It would be more convincing to also show results on more challenging high-resolution datasets (e.g. LSUN, ImageNet, CelebA) using pre-trained models, since the inference time of high-resolution models are ofter longer, and improving the inference time of larger models is more crucial.
4. The samples on ImageNet 64 in the appendix are not very good compared to the samples on CIFAR-10 in figure 2, which also raises questions on how general and robust the proposed approach is.
5. Given the existing work (e.g., DDIM), conditioned on all the previous noisier images in the sampling chain might not be very novel.

---

> ### Author Response · Authors · 2021-11-23
> **Reply to reviewer u4ae**
>
> Thank you for your valuable review. We address some of your specific concerns below.
>
> > It seems that optimizing the perceptual loss might not give you the samples from the true underlying DDPM model. Although this approach might produce reasonable samples empirically, it lacks theoretical insights: it is unclear what distribution these samples are optimized to match with. It seems that the training objective is hand-crafted, and might lack theoretical guarantees.
>
> Please see our general reply to all reviewers and the updated paper, where we now optimize for the Kernel Inception Distance (KID), as our original perceptual loss was almost identical to KID with the simplest (linear) kernel. With this loss, it is clear that we’re minimizing an unbiased estimator of the squared maximum mean discrepancy (MMD) between the data and model distributions.
>
> > Although samples from the proposed method have good FID/IS scores, optimizing the model via the perceptual loss could be directly/indirectly overfitting the evaluation metrics. So the improvements in FID/IS scores might be expected.
>
> Considering the very few degrees of freedom we are optimizing, it is very unlikely that we are overfitting. We will nevertheless include another ablation in the camera-ready version where we use different perceptual features for our KID loss.
>
> > Are you still able to get likelihoods (lower bound) on the generated samples from GGDP?
>
> Yes. We have updated the paper to now explicitly state the reverse process and the ELBO in Section 3. Still, we do not focus on likelihood as a *metric*, considering that multiple prior works in diffusion models have shown that log-likelihood and sample quality can be quite decorrelated to each other (see [Nichol & Dhariwal (2021)](https://arxiv.org/abs/2102.09672), [Watson et al. (2021)](https://arxiv.org/abs/2106.03802), [Song et al. (2021)](https://arxiv.org/abs/2101.09258)). This is also one of the main motivating factor for introducing a perceptual loss as opposed to optimizing for the ELBO directly!
>
> > For theorem 1, there is only "proof sketch". Is there a formal proof?
>
> We only phrased the proof as “proof sketch” because we did not redact the induction argument formally. We have now included it and moved the complete, formal proof to the appendix.
>
> > In section 4.1, it would be better to mention clearly what "the collection of all the introduced trainable variables" is.
>
> We have provided a more comprehensive level of detail on the parameterizations we optimize over, as this concern was also shared by reviewer vLgy. Please see the revised version of the paper!

---

> > ### Author Response · Authors · 2021-11-29
> > **Re. Reply to reviewer u4ae**
> >
> > Thanks again for your valuable review. We believe we should have addressed most of your comments in the revised version of the paper and in our responses. Please let us know if you have any other questions, comments or concerns!

---

### Official Review · Reviewer_zfCn · 2021-11-03

**Correctness:** 4
**Technical Novelty And Significance:** 3
**Empirical Novelty And Significance:** 3
**Recommendation:** 8
**Confidence:** 3

**Main Review:**

I find this paper to be very interesting and its results are quite good. I will address the main points of this paper as outlined above.

1. It appears that the introduction of a more flexible family of samplers is introduced mainly to admit the optimization process for finding the variances of a more efficient sampler (DDSS). As optimizing for $v_1 \dots v_k, \sigma$ would be difficult if the corresponding marginals were constrained to be equal. Indeed this is shown in 5.1 where optimizing $\sigma$ for DDIM does not perform well. This is to say that I am not sure there is any benefit or purpose to GGDP beyond admitting the use of DDSS. This is not necessarily a bad thing, but it's worth pointing out that GGPD doesn't look to stand on its own as a better sampler without DDSS. If this is not the case I am happy to be wrong here.

2. I'm not sure about the practicality of DDSS. Given that gradient rematerialization (only feasible in JAX as far as I know) is leveraged to compute the gradient of the objective, I can only assume that the computational cost of a naive implementation is quite high even for CIFAR10. That being said, this work is about efficient (in number of steps) samplers, not scalable ones. I found the results of GGDP+DDSS to be quite impressive. The samples shown in figure 2 are in my opinion strikingly better than the baseline approach.

I found the evaluation to be reasonable if a bit light on the number of datasets. Regardless, the results are compelling, and given that the DDSS objective is quite simple, I am happy with the approach.

All things considered, I think this optimization approach is a natural direction when considering how to construct more efficient DDPMs, and the results support it.

-------------------------------------------------------------
### Post Rebuttal Comments.

I appreciate the updates to the submission made by the authors. In particular, the improved motivation of using a perceptual loss instead of maximizing an ELBO. I will retain my score, though I hope that the authors follow through on their claim that they will provide additional results in the camera ready submission.


**Summary Of The Paper:**

This work examines the problem of the large number of iterations necessary to get high quality samples from denoising diffusion generative models.
The paper mostly consists of two new components:

1. A generalized family of probabilistic samplers called Generalized Gaussian Diffusion Processes (GGDP). This construction is similar to the denoising implicit processes found in [1], in the sense that it discards the Markov assumption, and $p(x_{t=i})$ may rely on all $p(x_{t < i})$. The difference is that the Gaussian marginals of the GGPD may not match those of the original process.

2. An optimization procedure for finding the sampler variances. This procedure is general and may be applied to any family of samplers. The idea is to minimize the distance in feature space between a classifier's representation given the real data, and the final generated iterate in the process, optimizing over the parameters of the sampler family. A computationally amenable scheme is proposed to allow for backpropagating through the diffusion process making use of gradient rematerialization (recomputation of intermediate gradients).

The method is compared against DDPMs [2] and DDIM [1] in CIFAR10 and Imagenet 64x64.

[1] Jiaming Song, Chenlin Meng, and Stefano Ermon. Denoising diffusion implicit models. arXiv preprint arXiv:2010.02502, 2020

[2] Jascha Sohl-Dickstein, Eric Weiss, Niru Maheswaranathan, and Surya Ganguli. Deep unsupervised learning using nonequilibrium thermodynamics. In International Conference on Machine Learning, pp. 2256–2265. PMLR, 2015


**Summary Of The Review:**

This work proposed a relatively simple, if computationally difficult optimization procedure for finding the parameters of a sampler able to generate compelling images in fewer iterations. I found the approach reasonable given that a more flexible family of samplers is proposed to be amenable to the optimization process. The results are compelling, achieving better sample quality in fewer timesteps. In the regime of fewer than 10 steps, this work achieves better sample quality in half the steps.

---

> ### Author Response · Authors · 2021-11-23
> **Reply to reviewer zfCn**
>
> Thank you for your valuable review. We agree that the empirical results show that DDPMs can be adapted for much faster sampling and better quality retention with our procedure, making DDPMs better suited for practical use. We address some of your concerns below:
>
> > I'm not sure about the practicality of DDSS. Given that gradient rematerialization (only feasible in JAX as far as I know) is leveraged to compute the gradient of the objective, I can only assume that the computational cost of a naive implementation is quite high even for CIFAR10.
>
> The objective is indeed slower to optimize than optimizing a DDPM, but with as few as 25,000 training steps the gains in sample quality are quite significant (outperforming DDIMs by a wide margin). In contrast, DDPMs typically require millions of training steps.
>
> > I am not sure there is any benefit or purpose to GGDP beyond admitting the use of DDSS.
>
> We agree an exciting direction for future work would be to try training VDMs (Variational Diffusion Models, Kingma et al., 2021) with more generalized families of forward processes like DDIM or GGDM, though this is beyond the scope of this paper, where the goal is to specifically show how to adapt DDPMs for faster sampling. One potential use is to further reduce the overall log-likelihood of DDPMs with the same model capacity (independently of the number of inference steps, e.g., the infinite-step limit ELBO in a VDM), which perhaps can lead to even better sample quality.

---

### Official Review · Reviewer_owpU · 2021-11-06

**Correctness:** 3
**Technical Novelty And Significance:** 3
**Empirical Novelty And Significance:** 3
**Recommendation:** 6
**Confidence:** 4

**Main Review:**

**The strengths of the paper**
In general, I found that the paper provides interesting values to the ML community:

1. It tackles an important problem in diffusion-based generative models, i.e., preserving the qualities of the generated samples with fewer discretization steps.
2. The paper proposes a new method using non-Markovian hierarchical generative models with the pre-trained diffusion-based generative models.
3. The paper discusses practical techniques to support the scalability of the proposed method.


**The weaknesses of the paper**
While the novelty of the proposed method, I found that the following aspects can be improved.

First of all, the motivation of the proposed model needs to be clarified, meaning the authors need to discuss the optimality of the non-Markovian but simple Gaussian hierarchical generative models and the proposed posteriors. I also consider this issue connects to the insufficient justification of minimizing the conceptual loss instead of maximizing the ELBO.

- For example, suppose the proposed model performs significantly worse when it is trained via maximizing ELBOs. In that case, it implies that the proposed posteriors are not well suited for the proposed generative models. This means the ELBO can not be tightened regardless of the number of discretization steps. Consequently, the pair of the proposed posterior and the generative model are inevitably suboptimal, similar to simple variational autoencoders. This raises the question of what family of densities the proposed model can model or not.

- On the contrary, DDIM is designed under the premise that the DDIM models will follow the same marginal $p(x_t)$ of the DDPM models. This implies that DDPMs' expressivity and tightness of the ELBO indirectly justify DDIMs'. Thus, DDIMs' approach makes much more sense, as we can expect that DDIM will model arbitrary densities by maximizing their ELBOs (or weighted ones). Thus, the tuning weights for DDIMs and DDPMs are valid engineering trade-offs depending on the application scenarios.

    - A side note: Sohl-Dickstein et al., 2015 and DDPMs are inspired by the fact that the first-order discretization of continuous-time It\^o diffusion is a Markov chain, each of whose transition probability follows a Normal distribution. Moreover, it has been proven that their reverse-time It\^ diffusion process exists (under mild conditions), and thus their first-order discretization again is Gaussian Markov chains.

- To resolve this, I consider that it is necessary to either provide (1) theoretical proofs that the proposed generative models are arbitrary expressive and the ELBO with the posteriors can be tightened, or (2) empirical analyses that the proposed method can achieve comparable performance when it is trained via maximizing ELBOs. Unless either of them cannot be demonstrated, the significance of the proposed method is doubtful.

Second, the introduction of the proposed method can be improved. For instance, the authors briefly mention that the proposed method will use pre-trained reconstruction networks in the abstract and Sec 2. However, I found that clarifying how Equation 7 is used to define generative models will be helpful for readers.

Thirds, the descriptions of the proposed method in the context of the previous works seem confusing. For example, the usage of the term "diffusion process" sounds confusing. "Diffusion process" have commonly referred to as It\^o diffusion process, which is a solution of specific types of stochastic differential equations; thus, continuous-time models. I found that it is a little problematic to use the "diffusion process" for the proposed method.

**Summary Of The Paper:**

The paper aims at reducing the number of discretization steps for the generation process of diffusion-based generative models. To do that, the paper proposes a new probabilistic generative model, which exploits pre-trained denoising diffusion probabilistic models. In particular, similarly to Denoising Diffusion Implicit Models (DDIM, Song et al. (2020)), the paper proposes a non-markovian hierarchical generative model, whose densities are defined as;
- $p(x_0, ..., x_T) = \prod_{t =0}^{T} p(x_{t} | x_{>t})$, where $p(x_T)$ is a prior distribution such as a standard Normal distribution.
- $p(x_{t} | x_{>t}) = q_{\mu, \sigma}(x_{t} |x_{>t}, x_0 = \hat{x}(x_{t}, t) )$
- $q_{\mu, \sigma}(x_{t} |x_{>t}, x_0) = N(x_t | \Sigma_{u = t+1}^{T} \mu_{t, u} x_u, \sigma_t^2 I_d)$
- $\hat{x}(x_t, t)$ is a reconstruction network similar to the one in the DDIM.
- Note that the proposed method employs $q_{\mu, \sigma}(x_{t} |x_{>t}, x_0))$, while DDIM uses conditionally Markov $q_{\mu, \sigma}(x_{t} |x_{s}, x_0)$ for any $t < s$.

Noting that the discrete-time diffusion-based generative models are fully differentiable, the paper proposes to train the models while the pre-trained reconstruction network is fixed; here, the authors name this training as Differentiable Diffusion Sampler Search (DDSS). To train the models, the paper first emphasizes that maximizing the ELBO of the proposed method deteriorates the performance wrt FID. Then the authors instead propose to minimize perception scores between models' sample and training data. Here, the perception score is a squared distance between empirical means of two sets of samples on the feature space of a pre-trained classifier.

Moreover, observing that training the entire discrete-time diffusion-based generative models requires huge memory costs, the paper proposes to use gradient rematerialization (Kumar et al., 2019b) methods.

In the experiments, the paper tests the effectiveness of the proposed method in image generation quality benchmarks, such as CIFAR-10. Specifically, the analysis of the experiments focuses on two perspectives. First, the authors compare the generation qualities of the proposed method and the baseline methods, including denoising diffusion probabilistic models (DDPM), in the few-step regime. Second, the paper compares the generation qualities of the DDIM and the proposed model when both are trained via DDSS. The paper claims that the proposed generative models trained via DDSM outperform the previous methods in the few-step regime.

===== POST-REBUTTAL COMMENTS ========
In the updated version, Theorem 1 shows that under certain choices of model parameterizations, the proposed models' ELBOs will be the same as the DDPMs'. This clarifies that it is valid to plug-in pre-trained DDPM models (or other similar types), potentially without fine-tuning; therefore, it seems valid to treat the proposed method as improving sampling process.

**Summary Of The Review:**

In general, the paper's contributions are unclear, while the paper tackles a very interesting problem in the context of diffusion-based generative models and provides empirical improvements. Unfortunately, the proposed method is not well-motivated, and the discussion about it is insufficient. Thus, I opt to reject the paper. However, I'm also inclined to improve my evaluation if the aforementioned weak points are well addressed.

===== POST-REBUTTAL COMMENTS ========
The rebuttal had addressed some of my concerns. Consequently, I raised my score from 3 to 6, and I also raised the "Correctness" & "Technical Novelty And Significance" scores from 2 to 3.

---

> ### Author Response · Authors · 2021-11-23
> **Reply to reviewer owpU**
>
> Thank you for your valuable review and feedback. We agree with all of the points in this review regarding the clarity and have incorporated the changes– please see the revised version of the paper. We now address your key concern:
>
> > In general, the paper's contributions are unclear, while the paper tackles a very interesting problem in the context of diffusion-based generative models and provides empirical improvements. Unfortunately, the proposed method is not well-motivated, and the discussion about it is insufficient. Thus, I opt to reject the paper. However, I'm also inclined to improve my evaluation if the aforementioned weak points are well addressed.
>
> We completely agree with you that, in our original submission, our loss function was not sufficiently well-motivated. We have now updated the paper, establishing a clear connection between our perceptual loss and the Kernel Inception Distance. Now, it is much more clear that we’re matching the sampler’s distribution with the data distribution, which hopefully should resolve some of the motivation concerns.
>
> As to *why* we choose to optimize KID (or more generally, a perceptual loss) instead of log-likelihood:
>
> As mentioned in the paper, there are several prior papers that motivate this choice. For instance, [Nichol & Dhariwal (2021)](https://arxiv.org/abs/2102.09672), [Watson et al. (2021)](https://arxiv.org/abs/2106.03802), [Song et al. (2021)](https://arxiv.org/abs/2101.09258) show that improvements in log-likelihood actually result in worse FID scores and sample quality. Even the seminal DDPM paper ([Ho et al., 2020](https://arxiv.org/abs/2006.11239)) shows that optimizing the exact variational lower bound results in lower sampled quality compared to their $L_{\mathrm{simple}}$ objective. **The goal of our paper is specifically to address this issue of low sample quality in the context of fast DDPM sampling.**
>
> If the reviewer feels that the motivation of either (1) our revised perceptual loss (KID) or (2) for *not* optimizing ELBO is insufficiently clear in the paper, we are more than happy to address this further.
>
> - - -
>
> As to some of your other concerns:
>
> > DDIM is designed under the premise that the DDIM models will follow the same marginal of the DDPM models. This implies that DDPMs' expressivity and tightness of the ELBO indirectly justify DDIMs'. Thus, DDIMs' approach makes much more sense, as we can expect that DDIM will model arbitrary densities by maximizing their ELBOs (or weighted ones).
>
> Note that the DDIM authors did not optimize their family of models at all. Rather, they empirically show that the choice of 0 variances simply has a tendency to work better for few-step diffusion sampling. Please also see the revised version of the paper. We provide more results showing that optimizing the reverse process variances with DDSS, *without* adjusting the mean coefficients following DDIM (i.e., not enforcing the marginals to match), leads to better results than both the DDIM($\eta=0$) baseline and DDSS applied to the DDIM family. This result suggests that enforcing this condition is likely not helpful in practice. Also note that our optimization family includes DDIM as a special case, so we strictly have more expressivity, not less.
>
> >[...] the introduction of the proposed method can be improved. For instance, the authors briefly mention that the proposed method will use pre-trained reconstruction networks in the abstract and Sec 2. However, I found that clarifying how Equation 7 is used to define generative models will be helpful for readers.
>
> Thank you for the feedback. We have updated the paper to include these improvements.
>
> >[...] the descriptions of the proposed method in the context of the previous works seem confusing. For example, the usage of the term "diffusion process" sounds confusing.
>
> Thanks, we agree that the term diffusion process can be confusing. Hence, we have renamed “Generalized Gaussian Diffusion Processes” to “Generalized Gaussian Diffusion *Models*” throughout the paper. This makes it easier for the reader to identify DDPM, DDIM, GGDM as different families of diffusion models, and does not sound like a mathematical object like an Ito diffusion.

---

> > ### Comment · Reviewer_owpU · 2021-11-29
> > **Reviewer response**
> >
> > Apologize for the late response, and thank you for the updates.
> >
> >
> > I find it might be helpful to clarify one point that I made in my review, **"To resolve this, I consider that it is necessary to either provide (1) theoretical proofs ..."**
> >
> >
> > In the original submission, it is unclear how the proposed method is different from a scenario where the pre-trained DDPMs are used as initializations of arbitrary generative models and trained in adversarial training (or minimizing KIDs). This is particularly important to clarify since the paper proposes minimizing KID, not maximizing the ELBOs.
> >
> >
> > Regarding this, the updated Theorem 1 has resolved some. In particular, Theorem 1 shows that under certain choices of model parameterizations, the proposed models' ELBOs will be the same as the DDPMs'. The theorem further clarifies that it is valid to plug in pre-trained DDPM models (or other similar types), potentially without fine-tuning; therefore, treating the proposed method as an improving sampling process seems valid. I consider this is a significant improvement of the current version from the initial submission.
> >
> >
> > Considering the goal of the proposed method is to improve FID scores (not ELBOs) with fewer sampling steps, it is reasonable to choose a differentiable objective function that correlates with the FID score. Hyperparameter searches of weights/coefficients won't be feasible anyway.
> >
> >
> > On the contrary, it is still unclear how to interpret the proposed models and the experimental results. Acknowledging that the KID uses Inception features (more precisely feature extractors trained from ImageNet datasets) and the FID scores use the same, it is straightforward to see that minimizing this loss will improve the FID scores of the proposed models. Nevertheless, one may ask some questions about the significance of the paper. For instance, is the improvement from minimizing KID? or due to the non-Markovian structure of the proposed models? Table 4 partially answers this. Another question may include whether the updated coefficients after KID minimization will deteriorate ELBOs and how different they are. Analyzing these questions may reveal to us what these models do and what improvements the models have made.
> >
> >
> > While the authors had not addressed this concern explicitly, the updated version has addressed some. Consequently, I will raise my score from 3 to 6.

---

### Author Response · Authors · 2021-11-23
**Reply to all reviewers**

Dear reviewers,

Thank you for your valuable feedback! We have updated the paper to address your concerns. We would appreciate it if you could take another look. We highlight the following changes:

0. We have renamed our sampler family from “Generalized Gaussian Diffusion Processes” (GGDP) to “Generalized Gaussian Diffusion Models” (GGDM) in the paper (kudos to reviewer owpU for the suggestion).

1. Sharing the concern that the original perceptual loss was not sufficiently well-motivated and theoretically grounded, we note that this loss is *almost* identical to the Kernel Inception Distance (KID) ([Bińkowski et al., 2018](https://arxiv.org/abs/1801.01401)) with a linear kernel $k(z,z’)=z^\top z’$ to compare inception features $z$ and $z’$. The only difference between the KID loss and our original loss is the exclusion of terms $i\neq j$ in the left summation term: $$\frac{1}{n(n-1)}\sum_{i\neq j}^n \left(k(z_p^{(i)},z_p^{(j)}) +  k(z_q^{(i)},z_q^{(j)})\right) - \frac{2}{n^2}\sum_{i=1}^n\sum_{j=1}^n k(z_p^{(i)},z_q^{(j)})~,$$
where  $z_q^{(i)}$ and $z_p^{(i)}$ denote inception features for real and synthesized samples for $i=1,...,n$. The exclusion of these terms results in an unbiased estimator of the squared maximum mean discrepancy (MMD). **We also re-ran all our experiments with this KID-based perceptual loss.** The sample quality scores we get are extremely similar to our original results, but now it is much more clear that distribution matching is happening with this choice of perceptual loss.

2. Following the KID paper, we also tried a cubic kernel  $k(z,z’)=\left(\frac{1}{d}z^\top z’+1\right)^3$. Surprisingly, the linear kernel performs very slightly better, despite that the cubic kernel should match more moments. Still, the similarity of the scores across sample quality metrics suggests our method has robustness to the choice of kernel. Results are included in the appendix.

3. To better motivate our choice of not enforcing the marginals of GGDM to match that of the original DDPM, we provide more empirical evidence showing matching the marginals is not helpful. Namely, we apply DDSS to optimize the variance coefficients of a DDPM, without adjusting the mean coefficients as done by a DDIM to make the marginals match. We find that this mostly outperforms both the strongest DDIM($\eta=0$) baseline and DDSS applied to the DDIM-sigmas family.

4. We incorporated the feedback regarding the clarity of the exposition. We believe the new version of the paper is clearer, thanks to your feedback.

Finally, while the paper already includes CIFAR and ImageNet experiments (ImageNet is especially challenging due to its large diversity and size), we will include results on another challenging dataset for the camera-ready version.

---

> ### Author Response · Authors · 2021-11-26
> **Re. Reply to all reviewers**
>
> Thank you once again for your valuable feedback. We believe we have addressed most of the concerns related to clarity in the revised version of the paper, and more importantly, the concerns related to the theoretical groundedness and motivation of our methodology. Please let us know if you have any other questions or comments; we are more than happy to address them before the end of the discussion period!

---

### Decision · Program_Chairs · 2022-01-20

**Decision:**

Accept (Poster)

**Comment:**

The paper tackles a very interesting problem in the context of diffusion-based generative models and provides empirical improvements. Pre-rebuttal, reviewers' main concerns lie in the motivation and clarification of the method, while after rebuttal, all reviewers satisfied the response and gave positive scores. The authors should include the additional results to well address the reviewers' concerns in the final version.